# A Sustainable Deep Learning Framework for Object Recognition Using Multi-Layers Deep Features Fusion and Selection

**Muhammad Rashid [1], Muhammad Attique Khan [1,\*], Majed Alhaisoni [2], Shui-Hua Wang [3,\*], Syed Rameez Naqvi [4], Amjad Rehman [5] and Tanzila Saba [5]**

1   Department of Computer Science, HITEC University Taxila, Taxila 47080, Pakistan; rashidraocomsats@gmail.com
2   College of Computer Science and Engineering, University of Ha'il, Ha'il 55211, Saudi Arabia; majed.alhaisoni@gmail.com
3   School of Architecture Building and Civil Engineering, Loughborough University, Loughborough LE11 3TU, UK
4   Department of EE, COMSATS University Islamabad, Wah Campus, Wah 47040, Pakistan; rameeznaqvi@ciitwah.edu.pk
5   College of Computer and Information Sciences, Prince Sultan University, Riyadh 11586, Saudi Arabia; drrehman70@gmail.com (A.R.); tsaba@psu.edu.sa (T.S.)
\*   Correspondence: attique@ciitwah.edu.pk (M.A.K.); shuihuawang@ieee.org (S.-H.W.)

**Abstract:** With an overwhelming increase in the demand of autonomous systems, especially in the applications related to intelligent robotics and visual surveillance, come stringent accuracy requirements for complex object recognition. A system that maintains its performance against a change in the object's nature is said to be sustainable and it has become a major area of research for the computer vision research community in the past few years. In this work, we present a sustainable deep learning architecture, which utilizes multi-layer deep features fusion and selection, for accurate object classification. The proposed approach comprises three steps: (1) By utilizing two deep learning architectures, Very Deep Convolutional Networks for Large-Scale Image Recognition and Inception V3, it extracts features based on transfer learning, (2) Fusion of all the extracted feature vectors is performed by means of a parallel maximum covariance approach, and (3) The best features are selected using Multi Logistic Regression controlled Entropy-Variances method. For verification of the robust selected features, the Ensemble Learning method named Subspace Discriminant Analysis is utilized as a fitness function. The experimental process is conducted using four publicly available datasets, including Caltech-101, Birds database, Butterflies database and CIFAR-100, and a ten-fold validation process which yields the best accuracies of 95.5%, 100%, 98%, and 68.80% for the datasets respectively. Based on the detailed statistical analysis and comparison with the existing methods, the proposed selection method gives significantly more accuracy. Moreover, the computational time of the proposed selection method is better for real-time implementation.

**Keywords:** object classification; deep learning; features fusion; features selection; recognition

## 1. Introduction

Object recognition is currently one of the most focused areas of research due to its emerging application in intelligent robotics and visual surveillance [1,2]. The researchers, however, are still facing problems in this domain for correct object recognition, such as in recognizing an object's shape and spotting a minor difference among several objects. Therefore, a sustainable system—the one

that maintains its performance against a change in the object's nature—is required for the correct recognition of complex objects [3]. Object classification is the key to a sustainable visual surveillance system [4]. Besides the latter, object classification finds its application in numerous domains, including intelligent robotics, face and action recognition, video watermarking, pedestrian tracking, autonomous vehicles, semantic scene analysis, content-based image retrieval, and many more. We believe that a genuinely sustainable object recognition system still has to overcome numerous challenges, including complex background, different shape and same color for different objects, continuously moving objects, different angles, and many more, since the conventionally used—unsustainable systems—did not prove to work well for complex object classification [5].

Many techniques have been introduced in computer vision to overcome the previously discussed challenges related to complex objects. What most of them tried to accomplish was an optimal method that would perform the same for many types of problems, but this was a considerable challenge. Although in the past few decades, the conventional approaches, such as Hand-Crafted Features (HCF), were used, as the time passed, however, the objects and their backgrounds became more confusing, thereby restricting their use. Handcrafted features included Histogram of Oriented Graph (HOG) [6], geometric features [7], Scale Invariant Feature Transformation (SIFT) [8], Difference of Gaussian (DoG) [9], Speeded-Up Robust Features (SURF) [10], and texture features (HARLICK) [11]. Recent techniques, in contrast, proposed to exploit a hybrid set of features to get a better representation of an object [12]. Unfortunately, those techniques were unable to recognize the growing complexities of objects and images as well.

In the face of the challenges as mentioned earlier, the concept of deep learning has been recently introduced in this context, which has also shown improved performance against reduced computational time. With this, a large number of convolutional neural networks (CNN) pre-trained models have been proposed. This includes AlexNet [13], VGG (VGG-16, VGG-19 [14], GoogleNet [14], ResNet (Resnet-50, ResNet-102, and ResNet-152) [15], and Inception [16]; all these models are trained on the ImageNet dataset. Even with these contributions, however, acceptable accuracy has been difficult to achieve. This has given rise to the concept of features fusion [15,16]—a process of combining several feature populations into a single feature space, which has been adopted in various applications ranging from medical imaging to object classification [17–19]. The concept of features fusion does manage to achieve increased classification accuracy, but only at an increased computational cost. In addition, some of the recent works have shown that the fusion process may add irrelevant features that are not important for the classification task [17,18]. We believe that if the irrelevant features were selected and removed from the fused vector, then the computational time could be minimized with an increased accuracy.

Feature selection can be categorized into three: Filter-based, wrapper-based, and embedded. The filter-based selection selects the features from subsets independently. The wrapper-based methods initially assume the features, and then select them based on predictive power. The embedded selection initially utilizes the selection in the training phase, which enjoys the advantages of both filter-based and wrapper based [19]. Some of the famous feature selection techniques include Principle Component Analysis (PCA) [20], Linear Discriminant Analysis (LDA) [21], Pearson Correlation Coefficient (PCC) [22], Independent Component Analysis (ICA) [22], Entropy Controlled [23], Genetic Algorithm-based (GA) [24], and many more.

In this work, an entire sustainable framework based on a deep learning architecture is proposed. While we summarize our challenges and highlight our contributions in response to those in Section 3, the details on the proposed framework are explicitly given in Section 4. Section 5 presents the simulation results before we conclude the manuscript in Section 6. In what follows, however, we review some of the existing related works, in Section 2.

## 2. Related Work

Many strategies are performed for image classification, as investigated in the area of computer vision and machine learning. Object categorization is the most emergent field of computer vision

because of its enormous applications in video surveillance, auto-assisted vehicle frameworks, pedestrian analysis, automatic target recognition, and so on. In the literature, very few fusion-based techniques are presented for the classification of complex objects. Features fusion is the process of combining two or more feature spaces into a single matrix. By fusion, there is a chance to get a higher accuracy vector having properties of multiple feature spaces. Roshan et al. [25] presented a new technique for object classification. They applied the presented algorithm on the VGG-16 architecture and performed training from scratch. Additionally, they used transfer learning (TL) on the top layers. They utilized the Caltech-101 dataset and achieved an accuracy of 91.66%. Jongbin et al. [26] introduced a new DFT-based technique for feature building by discarding the pooling layers among the fully connected and convolutional layers. Two modules were implemented in this technique: The first module, known as DFT, replaced max-pooling from the architecture by a user-defined size pooling. The second module, known as DFT+, was the fusion of multiple layers to get the best classification accuracy. They achieved 93.2% classification accuracy on the Caltech-101 dataset using the VGG-16 CNN network, and 93.6% accuracy on the same dataset using the Resnet-50 model. Qun et al. [27] used a pre-trained network with associative memory banks for feature extraction. They extracted the features using ResNet-50 and VGG-16. Later, the K-Means clustering was used on the memory banks to perform unsupervised clustering. Qing et al. [28] presented a fused framework for object classification. They extracted the CNN features and applied three different types of coding techniques onto the fused vector.

Two pre-trained models, namely VGG-M and VGG-16, were used for feature extraction from the 5-Conv-Layer. Subsequently, PCA-based reduction was applied, and features were fused into a final vector using the proposed coding techniques. Results showed an improved accuracy of 92.54% on the Caltech-101 database. Xueliang et al. [29] presented a late fusion-based technique for object recognition. Three pre-trained networks, namely AlexNet, VGGNet, and ResNet-50, were used for the purpose. Firstly, they evaluated that the middle-level layers of the CNN architecture contained more robust information for visual representation, and then features were extracted from these layers. Features fusion from these three models showed the improved result, and reported 92.2% accuracy on the Caltech-101 dataset. Hamayun et al. [30] proved that the most robust features were extracted from the fully connected layer-6 (FC-6) instead of the FC-8. In the presented approach, they exploited the CNN output and modified it at a middle-level layer instead of the deepest layer. VGG-16 and VGG-19 pre-trained models were used to illustrate the proposed technique. They extracted 4096 features from the FC-6 layer and then applied reduction using PCA. For the experimental process, they used the Caltech-101 dataset and attained an accuracy of 91.35% using the reduced features from the layer FC-6. Mahmood et al. [31] gave an idea for object detection and classification using pre-trained networks (ResNet-50 and ResNet-152). After feature extraction, they performed features reduction using PCA. The Caltech-101 database was selected for evaluation and achieved an accuracy of 92.6%. Emine et al. [32] used convolutional architecture for fast feature embedding (Caffe) for object recognition. About 300 images from the Caltech-101 dataset were used to test the proposed technique. Results showed that 260 images were correctly classified, and 40 were misclassified. Chunjie et al. [33] introduced a new technique, called Contextual Exemplar, to handle the drawbacks caused by the local features. The method comprised three phases: In the first, they combined the regions-based image, followed by constructing the relationship between those regions in the second phase, and they used the connection of those regions for semantic representation in the third phase. They selected 1000 features and achieved an accuracy of 86.14%. Rashid et al. [8] focused on multiple features fusion and selection of the best of them for efficient object classification. They used VGG and Alexnet pre-trained models for CNN feature extraction and SIFT as point features extraction. Both types of features were fused by a simple concatenation approach. Moreover, they introduced an entropy-based selection approach within their framework, which achieved an accuracy of 89.7% for the Caltech-101 dataset. Nazar et al. [34] fused HOG and Inception V3 CNN features and improved the existing accuracy up to 90.1% for the Caltech-101 dataset.

## 3. Challenges and Contributions

The computer vision research community is still facing various challenges for object classification, and most of them are due to the complex nature of objects. We do realize that it is not an easy task to classify objects into their relevant categories efficiently. To be able to tackle the challenges facing the community and achieve the required accuracies, in this work, we propose a deep learning architecture-based framework for object classification with improved accuracy. The highlights of the framework are as follows:

- It uses two pre-trained deep learning architectures, namely-VGG19 and Inception V3, and performs TL to retrain the selected datasets. The FC7 and Average Pool layers of the CNN are utilized for feature extraction.
- A parallel maximum covariance (PMC) technique is proposed for the fusion of both deep learning feature vectors.
- While the Multi Logistic Regression controlled Entropy-Variances (MRcEV) method is employed for selecting the robust features, the Ensemble Subspace Discriminant (ESD) classifier is used as a fitness function.
- A detailed statistical analysis of the proposed method is conducted and compared with recent techniques to examine the stability of the proposed architecture.

## 4. Materials and Methods

The proposed object classification architecture is presented in this section with detailed mathematical formulation and visible results. As shown in Figure 1, the proposed architecture consists of three core steps: Deep learning feature extraction using TL, fusion of various model features, and selection of the robust features for final classification. In the classification step, the ESD classifier is used, and the performance is compared with other learning algorithms. The details of each step, depicted in this figure, are discussed below.

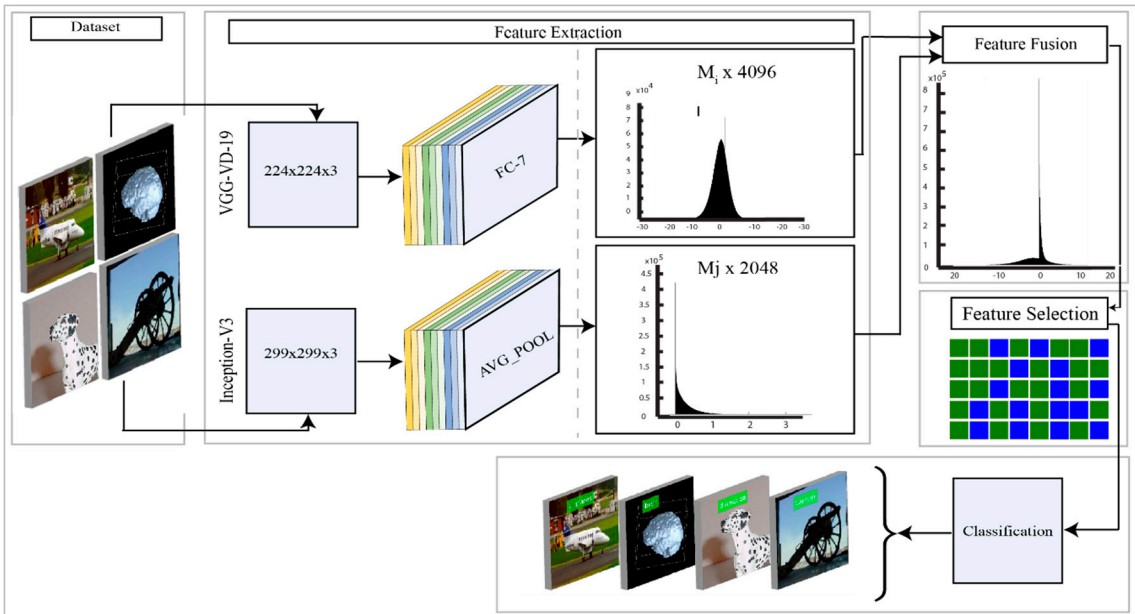

**Figure 1.** Proposed deep learning architecture for object classification.

### 4.1. Deep Learning Features Extraction

Since the past two decades, deep learning has proven itself as the best approach for image recognition and classification [8,35–37]. CNN is a method of deep learning, involving a series of layers. A simple CNN model consists of convolution and pooling layers. A few other layers are the

activation layer named ReLu, and the feature layer called fully connected (FC). The first layer of CNN is known as the input layer. This layer takes images as input, and the convolutional layer computes the neurons' response. The latter is calculated by the dot product of weights and smaller regions. While the ReLu layer helps in the activation function, the pooling layer between convolution layers removes the inactive neurons for the next phase. Finally, the high-level features are computed using the FC layers, which are classified through Softmax [8]. In this work, we are using two pre-trained CNN models, namely VGG19 and Inception V3, for feature extraction. In what follows, we present a brief description of each model.

**VGG19:** VGG-19 [38] consists of 16 convolutional layers, 19 learnable weights layers, which are utilized for transfer learning, 3 FC layers, and an output layer. This model is already trained on the ImageNet dataset. The input size for this model is $224 \times 224 \times 3$, as given in Table A1 (Appendix A). The learnable weights and bias of the first convolution layer are $3 \times 2 \times 3 \times 64$ and $1 \times 1 \times 64$. The total learnable at this layer is 1792. For the second convolution layer, the total learnable is 36,928. This layer extracts the local features of an image.

$$V_i^{(M)} = B_i^{(M)} + \sum_{k=1}^{n_1^{(M-1)}} \psi_{i,k}^{(M)} \times h_k^{(M-1)} \tag{1}$$

where, $V_i^{(M)}$ is the output layer $L_y$, $B_i^{(M)}$ is the base value, $\psi_{i,k}^{(M)}$ denotes the filter mapping the *kth* feature value, and $h_k$ means the $M-1$ output layer. The learnable weights and bias of the first FC layer are $4096 \times 25,088$ and $4096 \times 1$. The dropout layer is added between FC layers, where the dropout rate is 50%. For FC layer 7, the total learnable is 16,781,312, and learnable weights are $4096 \times 4096$. For the last FC layer, the total learnable is 4,097,000, and learnable weights are $1000 \times 4096$. Hence, when the activation is applied, it returns a feature map vector of dimension $1 \times 1 \times 1000$. For fully connected layers 1 and 2, the feature map vector dimension is $1 \times 1 \times 4096$.

**Inception V3**: It is an advanced pre-trained CNN model. It consists of 316 layers and 350 connections. The number of convolution layers is 94 of different filter sizes, where the size of the first input layer is $299 \times 299 \times 3$. A brief description of this model is given in Table A2 (Appendix A). In this table, it is shown that a scaling layer is added after the input layer. On the first convolution layer, activation is performed and obtained a weight matrix of dimension $149 \times 149 \times 32$, where 32 denotes the number of filters. Later, the batch normalization and ReLu activation layers are added. Mathematically, the ReLu layer is defined as:

$$\text{Re}_i^{(l)} = \max\left(hv, hv_i^{(l-1)}\right) \tag{2}$$

Between the convolution layers, a pooling layer is also added to get active neurons. In the first max-pooling layer, the filter size is $2 \times 2$. Mathematically, the max-pooling is defined as:

$$mx_1^{(q)} = mx_1^{(q-1)} \tag{3}$$

$$mx_2^{(q)} = \frac{mx_2^{(q-1)} - F(q)}{S^q} + 1 \tag{4}$$

$$mx_3^{(q)} = \frac{mx_3^{(q-1)} - F(q)}{S^q} + 1 \tag{5}$$

where, $S^M$ denotes the stride, $mx_1^M$, $mx_2^M$, and $mx_3^M$ are defined filters for feature set maps such as $2 \times 2$, $3 \times 3$. Moreover, a few other layers are also added in this architecture, such as addition and concatenation layers. In the end, an average pool layer is added. The activation is performed, and in the output, a resultant weight matrix is obtained as a features map of dimension $1 \times 1 \times 2048$. The last

layer is FC, and its learnable weight matrix is $1000 \times 2048$, and the ensuing feature matrix is $1 \times 1 \times 1000$. Mathematically, the FC layer is defined as follows:

$$Fc_i^{(l)} = f\left(z_i^{(l)}\right) with\, z_i^{(l)} = \sum_{j=1}^{n_1^{(l-1)}} \sum_{r=1}^{n_2^{(l-1)}} \sum_{s=1}^{n_3^{(l-1)}} w_{i,j,r,s}^{(l)} \left(Fc_i^{(l-1)}\right)_{r,s} \tag{6}$$

**Feature Extraction using TL**: In the feature extraction step, we employ TL, by which we retrain both the specific CNN models (VGG19 and InceptionV3) on the selected datasets. For training, we set a 60:40 approach along with labeled data. Furthermore, we perform preprocessing, in which we resize the images according to the input layer of each model. Later, we select the input convolutional and output layers as feature mapping. For VGG19, we choose the first convolutional layer as an input layer, and the FC7 as the output. After that, the CNN activation is performed, and we obtain the training and testing vectors. On the feature layer FC7, a resultant feature vector is obtained of dimension $1 \times 4096$ denoted by $\varphi^{(k1)}$ and utilized in the next process. A modified architecture of VGG19 is also shown in Figure 2. For Inception V3, we select the first convolutional layer as input, and the average pool layer as a feature map. Similar to VGG19, we perform TL and retrain this model on the selected datasets, and apply the CNN activation on the average pool layer. On this layer, we obtain a feature vector of dimension $1 \times 2048$, denoted by $\varphi^{(k2)}$. Both training and testing vectors proceed for the next features fusion process. The modified architecture of Inception V3 is shown in Figure 3. In this figure, it is shown that the last three layers are removed before being retrained on the selected datasets for this work.

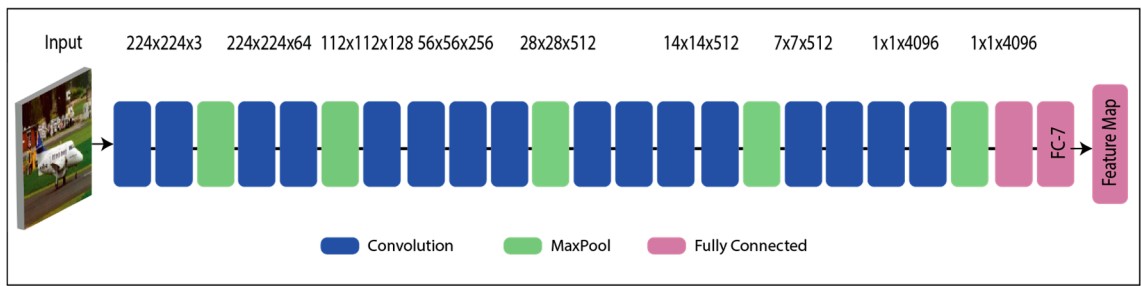

**Figure 2.** Modified VGG-19 architecture for features extraction.

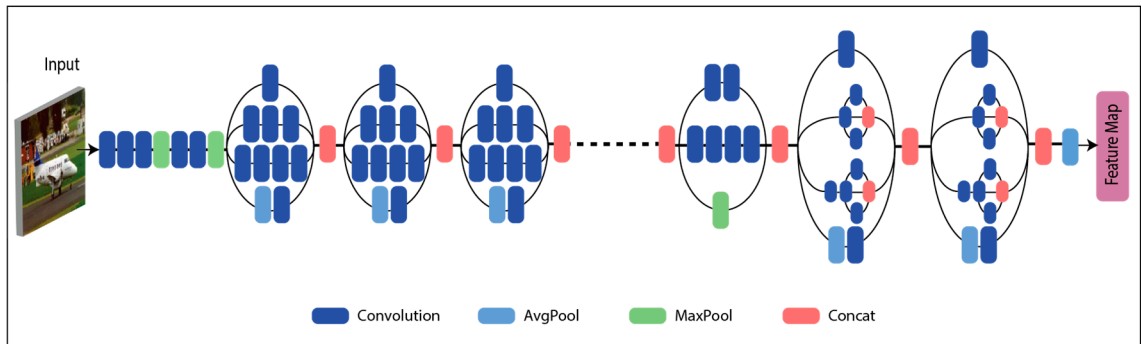

**Figure 3.** Modified Inceptionv3 architecture for features extraction.

### 4.2. Features Fusion

The fusion of multiple features in one matrix is the latest research area of pattern recognition. The primary purpose of features fusion is to obtain a stronger feature vector for classification. From the latest research, it is noticed that the fusion process improves the overall accuracy, but on the other side, its main disadvantage is high computational time (s). However, our usual priority is to improve the classification accuracy. For this purpose, we implement a new Parallel Maximum Covariance (PMC) approach for features fusion. In this approach, we need to equalize the lengths of both extracted feature vectors. Later, we find the maximum covariance for fusion in a single matrix.

Consider two deep learning feature vectors $\varphi^{(k1)}$ and $\varphi^{(k2)}$ of dimensions $n \times m$ and $n \times q$, where $n$ denotes the number of images, $m$ indicates VGG19 deep learning feature vector length of $n \times 4096$ and $q$ denotes Inception V3 feature vector of dimension $n \times 2048$, respectively. To make the length of vectors equal, we first find out the maximum length vector and perform average value padding. The average feature is calculated from a higher length vector. Let $a$ be an arbitrary unit column $m$ vector presenting a pattern in $\varphi_1$ field, and $b$ indicates a random unit column vector representing a pattern in the $\varphi_2$ field, respectively. The time series projections on row vectors are defined as follows:

$$x_1 = \varphi_1^T \, \varphi^{(k1)} \tag{7}$$

$$x_2 = \varphi_2^T \, \varphi^{(k2)} \tag{8}$$

For optimal solutions $\varphi_1$ and $\varphi_2$, maximize their covariance as follows:

$$\widetilde{c} = Cov[x_1, x_2] \tag{9}$$

$$\widetilde{c} = Cov[\varphi_1^T \, \varphi^{(k1)}, \varphi_2^T \, \varphi^{(k2)}] \tag{10}$$

$$\widetilde{c} = \frac{1}{n-1}[\varphi_1^T \, \varphi^{(k1)}(\varphi_2^T \, \varphi^{(k2)})] \tag{11}$$

$$\widetilde{c} = \varphi_1 \big( C_{\varphi_1 \varphi_2} \big) \varphi_2 \tag{12}$$

$$C_{\varphi_1 \varphi_2} = \frac{1}{n-1} \big( \varphi^{(k1)} \, \varphi^{(k2)T} \big) \tag{13}$$

where, $C_{\varphi_1 \varphi_2}$ is the covariance value among $\varphi_1$ and $\varphi_2$ whose $i$th and $j$th features are $\varphi_i(t)$ and $\varphi_j(t)$. Hence, the feature pair $i$ and $j$ of maximum covariance $C_{\varphi_1 \varphi_2}$ is saved in the final fused vector. However, it is possible that few of the feature pairs are redundant. This process is continued until all pairs are compared with each other. In the end, a fused vector is obtained, denoted by $\varphi^{(fu)}$ of dimensions $N \times K$, where $K$ denotes the feature-length, which varies according to the selected features. In this work, the fused feature-length is $N \times 3294$ for the Caltech-101 dataset, $N \times 2981$ for the Birds dataset, and $N \times 3089$ for the Butterflies dataset.

### 4.3. Feature Selection

Feature selection is an exciting research topic in machine learning (ML) nowadays, and shows significant improvement in the classification accuracy. In this work, we propose a new technique for feature selection, namely, Multi Logistic Regression controlled Entropy-Variances (MRcEV). It exploits a partial derivative-based activation function to remove the irrelevant features, and the remaining robust features are passed to the entropy-variances function. Through the latter, a new vector is obtained, which only contains positive values. Finally, this vector is presented to the ESD fitness function, and the validity of the proposed technique is determined. Mathematically, the formulation is given as:

For a given dataset, a fused vector is represented as $\Delta = \left\{ \varphi^{(fu)}, y^{(fu)} \right\}_{fu=1}^N$ having $N$ sample images, where $\varphi^{(fu)}$ denotes the fused feature vector, which is utilized as the input, and $\varphi^{(fu)} \in \mathbb{R}^p$. The $y^{(fu)}$ indicates the corresponding labels and defined as $y^{(fu)} \in \mathbb{R}$. The probability among $\varphi^{(fu)}$ for the class $i$ is then computed as follows:

$$p(y^{(fu)} | \varphi^{(fu)}) = \frac{\exp\left\{ r_i^{(fu)} \right\}}{\sum_{j=1}^q \exp\left\{ r_i^{(fu)} \right\}} \tag{14}$$

$$r_i^{(fu)} = \sum_{j=1}^p \beta_{ij} \varphi_j^{(fu)} \tag{15}$$

The parameter of logistic regression $r_i = (r_0, r_1, \ldots, r_p)$ is obtained by minimizing the negative likelihood of features. If features are independent, then a multinomial distribution is computed as follows:

$$E_\Delta = -\sum_{fu}^{n} \sum_{i=1}^{q} y_i^{(fu)} \log p(y^{(fu)} | \varphi^{(fu)}) \tag{16}$$

To get a sparse model, a regularization parameter $\widetilde{\beta}$ is added to negative log-likelihood. The modified MLR criteria for the active features are defined as follows:

$$M = E_\Delta + \widetilde{\beta} E_r \tag{17}$$

$$E_r = \sum_{i=1}^{p} |r_i| \tag{18}$$

where $r_i$ is regularization parameter.

At the minimum value of $M$, the partial derivative with respect to $r_i$ is formulated as follows:

$$\begin{cases} \left| \frac{\partial E_\Delta}{\partial r_i} \right| = \widetilde{\beta} & if \;\; |r_i| > 0 \\ \left| \frac{\partial E_\Delta}{\partial r_i} \right| < \widetilde{\beta} & if \;\; |r_i| = 0 \end{cases} \tag{19}$$

This expression shows that if the partial derivative of $E_\Delta$ with respect to $r_i$ is less then $\widetilde{\beta}$, then that feature value is set to zero, and removed from the final vector. Later, entropy-variances-based function is implemented to obtain a more robust vector. Mathematically, this function is formulated as:

$$H(\widetilde{\beta}) = -\sum_{i=0}^{N-1} p_i(\widetilde{\beta}) \log p_i(\widetilde{\beta}) \tag{20}$$

$$\sigma^2(\widetilde{\beta}) = \frac{\sum (\widetilde{\beta}_i - \overline{\overline{\beta}})}{n-1} \tag{21}$$

$$Ent(FV) = -\left( \frac{\ln(H(\widetilde{\beta})_{i+1}) + \sigma^2(\widetilde{\beta})}{\ln(H(\widetilde{\beta})_i + \sigma^2(\widetilde{\beta})) + \ln(H(\widetilde{\beta})_i - \sigma^2(\widetilde{\beta}))} \right) \tag{22}$$

where, $H(\widetilde{\beta})$ is an entropy function, $\sigma^2(\widetilde{\beta})$ denotes variance of the selected vector, and $Ent(FV)$ represents the final entropy-variances function. The selected features are passed to this function to get a clear difference among all the features based on the classification classes. This proposed selection technique picks almost 50% to 60% robust features from the fused feature vector. The selected features are finally verified through the ESD classifier [39]. In the ensemble learning classifier, the subspace discriminant method is used. The proposed system's predicted results are shown in Figures 4–6.

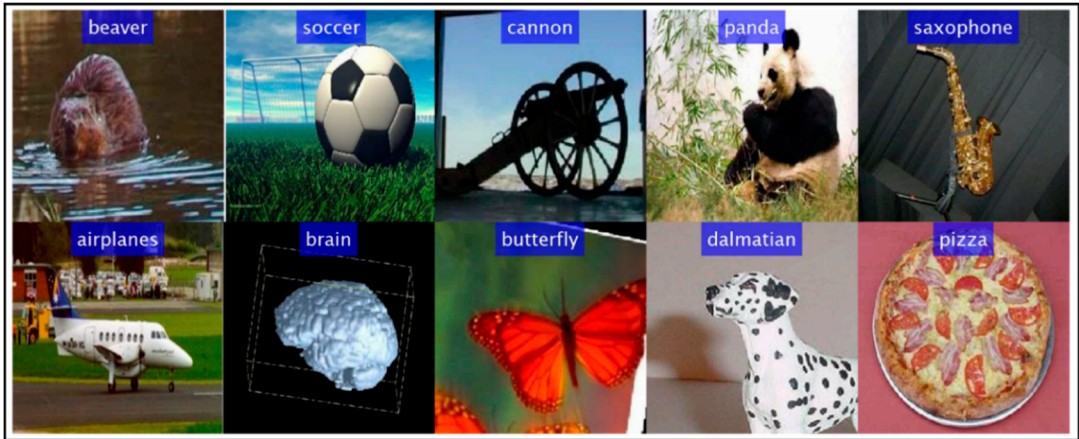

**Figure 4.** Proposed system's predicted labeled output for the Caltech-101 dataset.

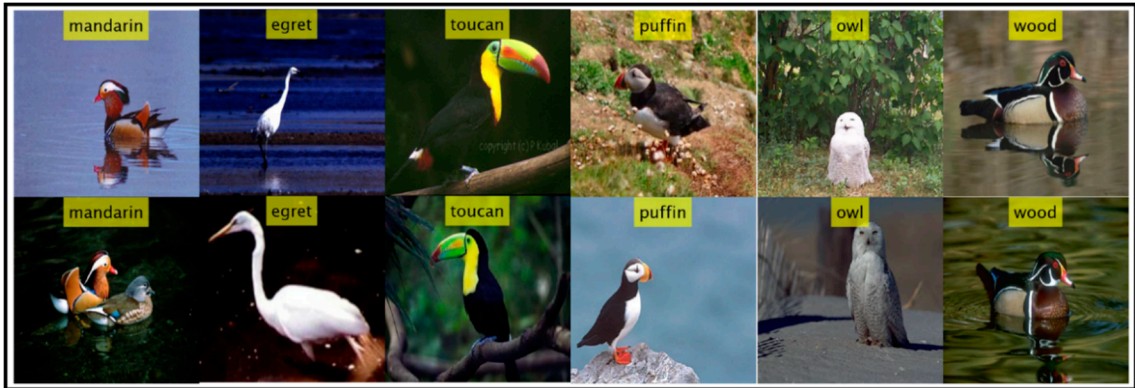

**Figure 5.** Proposed system's predicted labeled output for the Birds dataset.

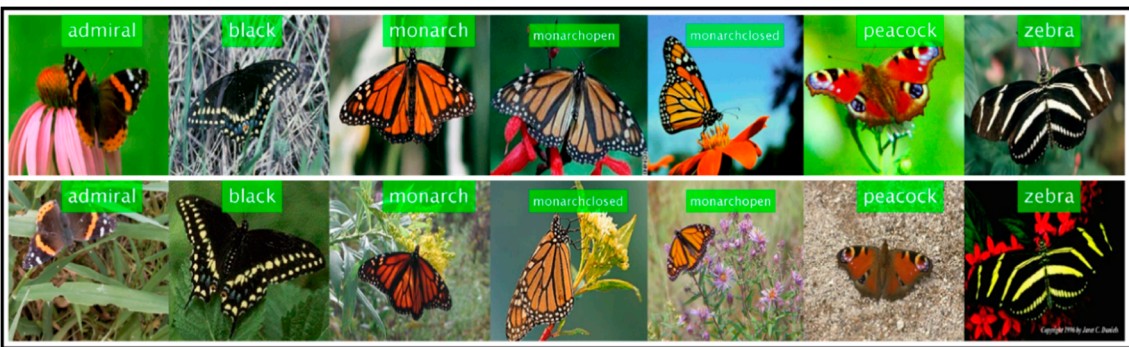

**Figure 6.** Proposed system's predicted labeled output for the Butterflies dataset.

## 5. Results

This section presents the simulation results with detailed numerical analysis and visual plots. As stated above, in this work, we utilize four publicly available datasets for evaluation of the proposed framework, including Caltech-101, Birds database, Butterflies database, and CIFAR-100 [40]. A brief description of the selected datasets is given in Table 1, where we have highlighted the total number of images, their specific classes (categories), and the number of images that each class comprises. As understandable, the Caltech-101 and CIFAR-100 are relatively more challenging for object classification. For validation, the 60:40 approach is employed along with ten-fold cross-validation. We used various classifiers for the experimental process, such as Ensemble learning, SVM, KNN, and Linear Discriminant classifiers. The performance of each classifier is validated using three essential measures, including accuracy, FNR, and computational time. All the simulations are conducted in

MATLAB2019a installed on a 2.4 Gigahertz Corei7 processor with 16 Gigabytes of RAM, 128 SSD, and a Radeon R7 graphic card.

**Table 1.** Numerical description of selected datasets.

| Image Database | Sample Classes | Total Samples | Min-Max |
|---|---|---|---|
| **Caltech** [41] | 101 | 9144 | 31~800 |
| **Birds** [42] | 6 | 600 | 100~100 |
| **Butterflies** [43] | 7 | 619 | 42~134 |
| **CIFAR-100** [40] | 100 | 1000 (Testing) 50,000 (Training) | 100 |

*5.1. Caltech-101 Dataset Results*

The results achieved on the Caltech-101 dataset are presented in three different ways: In the first method, both VGG19- and inceptionV3-based deep features are fused using a serial-based method, and the classification is performed without features selection. In the second method, the fusion of deep features is conducted using the proposed fusion approach, as presented in Section 4.2. In the third method, the feature selection is performed on the proposed fused vector, followed by classification. The results are shown in Table 2, where it is evident that the ESD classifier yields the best results against the rest for each method. However, it may be noticed that a massive difference exists among the accuracies achieved using M1 and the other methods. For example, consider the case of the ESD classifier, where the achieved accuracy rises from 79% to 90.8% upon using the proposed fusion method, which further jumps to 95.5% once the proposed selection method is applied. Additionally, observe that the computational time drops by around 74% between M1 and the P-selection method, making the latter more superior to the other two methods. The accuracy of the P-Selection method may also be verified through Figure 7. The effectiveness of the proposed P-Fusion and P-Selection methods while using other classifiers is also evident in Table 2. Observe that the best accuracies are provided by the P-Selection method irrespective of the classifier, while the P-Fusion stands second, both in terms of accuracy and computational time. Overall, the proposed selection method shows significant performance on ESD classifier for the Caltech-101 dataset.

**Table 2.** Proposed classification results using the Caltech-101 dataset. M1 represents simple serial-based fusion and classification, P-Fusion represents the proposed fusion approach, and P-Selection represents the proposed selection method results. Where, ESD described ensemble subspace discriminant, LDA represent linear discriminant analysis, LSVM denotes linear support vector machine, QSVM denotes quadratic SVM, and Co-KNN describe cosine K-Nearest Neighbor.

| Classifier | M1 | P-Fusion | P-Selection | Accuracy (%) | FNR (%) | Time (s) |
|---|---|---|---|---|---|---|
| | ✓ | - | - | **79.0** | **21.0** | 180.00 |
| **ESD** | - | ✓ | - | **90.8** | **9.2** | 93.70 |
| | - | - | ✓ | **95.5** | **4.5** | 47.00 |
| | ✓ | - | - | 75.8 | 24.2 | 665.80 |
| ES-KNN | - | ✓ | - | 80.1 | 19.9 | 286.45 |
| | - | - | ✓ | 85.3 | 14.7 | 191.27 |
| | ✓ | - | - | 75.0 | 25.0 | 597.84 |
| LDA | - | ✓ | - | 81.8 | 18.2 | 127.83 |
| | - | - | ✓ | 94.4 | 5.5 | 106.57 |

**Table 2.** *Cont.*

| Classifier | M1 | P-Fusion | P-Selection | Accuracy (%) | FNR (%) | Time (s) |
|---|---|---|---|---|---|---|
| | ✓ | - | - | 76.0 | 24.0 | 9723.70 |
| L-SVM | - | ✓ | - | 88.0 | 12.0 | 3154.70 |
| | - | - | ✓ | 91.6 | 8.6 | 2045.00 |
| | ✓ | - | - | 77.2 | 22.8 | 1896.00 |
| Q-SVM | - | ✓ | - | 87.6 | 12.4 | 1341.00 |
| | - | - | ✓ | 92.0 | 8.0 | 753.57 |
| | ✓ | - | - | 77.9 | 22.1 | 7493.00 |
| Cu-SVM | - | ✓ | - | 87.7 | 12.3 | 3647.70 |
| | - | - | ✓ | 92.3 | 7.7 | 1889.50 |
| | ✓ | - | - | 75.7 | 24.3 | 152.06 |
| F-KNN | - | ✓ | - | 84.9 | 15.1 | 96.96 |
| | - | - | ✓ | 89.9 | 10.1 | 71.57 |
| | ✓ | - | - | 74.8 | 25.2 | **57.95** |
| M-KNN | - | ✓ | - | 84.5 | 15.5 | 47.44 |
| | - | - | ✓ | 89.6 | 10.4 | 33.90 |
| | ✓ | - | - | 76.8 | 23.2 | 228.19 |
| W-KNN | - | ✓ | - | 85.7 | 14.3 | 187.50 |
| | - | - | ✓ | 90.5 | 9.5 | 105.87 |
| | ✓ | - | - | 52.4 | 21.0 | 61.35 |
| Co-KNN | - | ✓ | - | 87.6 | 12.4 | 48.76 |
| | - | - | ✓ | 92.8 | 7.2 | 23.83 |

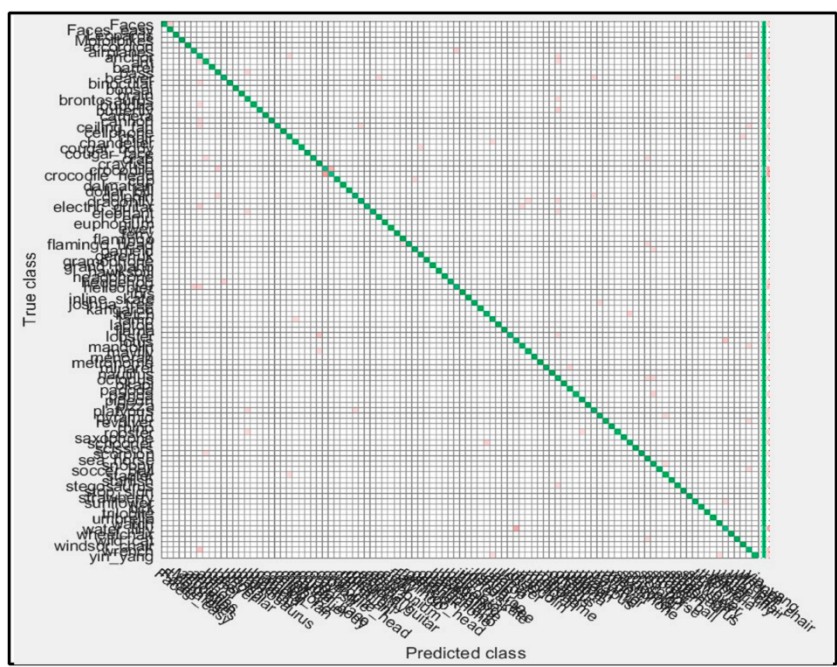

**Figure 7.** Confusion matrix of the proposed selection accuracy on ESD classifier.

## 5.2. Birds Dataset Results

The classification results using the Birds dataset are presented in this section. As before, three methods are applied for the evaluation, and all the results obtained previously hold true in this case as well. Table 3 summarizes these results, and verifies that the ESD classifier yields the best results for all the three methods when compared with various classifiers. Irrespective of the classifier used, it may also be verified that the proposed fusion method outperforms the M1 both in terms of the achieved accuracies and computational time, while the proposed selection method even surpasses the fusion method in both metrics. Its accuracy is also confirmed by Figure 8. Due to the simplicity in the dataset, the accuracies achieved by the three methods are relatively comparable, unlike in the case of Caltech-101, where the proposed methods outperformed the M1 by a considerable margin. The computational time, however, gives the proposed methods a substantial edge on the equivalent techniques.

**Table 3.** Proposed classification results using the Birds dataset. M1 represents simple serial-based fusion and classification, P-Fusion represents the proposed fusion approach, and P-Selection represents the proposed selection method results.

| Classifier | M1 | P-Fusion | P-Selection | Accuracy (%) | FNR (%) | Time (s) |
|---|---|---|---|---|---|---|
| | ✓ | - | - | **99.0** | 15.5 | 85.09 |
| ESD | - | ✓ | - | **99.5** | 1.0 | 68.31 |
| | - | - | ✓ | **100.0** | 0.0 | 42.45 |
| | ✓ | - | - | 96.7 | 3.3 | 45.09 |
| E-S-KNN | - | ✓ | - | 97.6 | 2.4 | 38.31 |
| | - | - | ✓ | 97.4 | 2.6 | 25.54 |
| | ✓ | - | - | 98.0 | 2.0 | 48.39 |
| LD | - | ✓ | - | 99.0 | 1.0 | 31.11 |
| | - | - | ✓ | 100.0 | 0.0 | 23.92 |
| | ✓ | - | - | 97.9 | 2.1 | 45.36 |
| L-SVM | - | ✓ | - | 99.0 | 0.5 | 20.00 |
| | - | - | ✓ | 100.0 | 0.0 | 17.66 |
| | ✓ | - | - | 84.5 | 1.0 | 51.03 |
| Q-SVM | - | ✓ | - | 99.3 | 0.7 | 24.06 |
| | - | - | ✓ | 100.0 | 0.0 | 15.25 |
| | ✓ | - | - | 99.0 | 1.0 | 54.59 |
| Cub-SVM | - | ✓ | - | 99.5 | 0.5 | 43.32 |
| | - | - | ✓ | 100.0 | 0.0 | 21.29 |
| | ✓ | - | - | 96.2 | 3.8 | 41.47 |
| F-KNN | - | ✓ | - | 97.4 | 2.6 | 19.58 |
| | - | - | ✓ | 99.5 | 0.5 | 14.89 |
| | ✓ | - | - | 97.6 | 2.4 | 32.30 |
| M-KNN | - | ✓ | - | 98.8 | 1.2 | 17.31 |
| | - | - | ✓ | 100.0 | 0.0 | 15.82 |
| | ✓ | - | - | 97.9 | 2.1 | 23.96 |
| W-KNN | - | ✓ | - | 99.3 | 0.7 | 13.10 |
| | - | - | ✓ | 100.0 | 0.0 | 9.16 |

**Table 3.** *Cont.*

| Classifier | M1 | P-Fusion | P-Selection | Accuracy (%) | FNR (%) | Time (s) |
|---|---|---|---|---|---|---|
| | ✓ | - | - | 95.7 | 4.3 | 31.08 |
| Cos-KNN | - | ✓ | - | 99.0 | 1.0 | 22.00 |
| | - | - | ✓ | 99.8 | 0.2 | 16.11 |

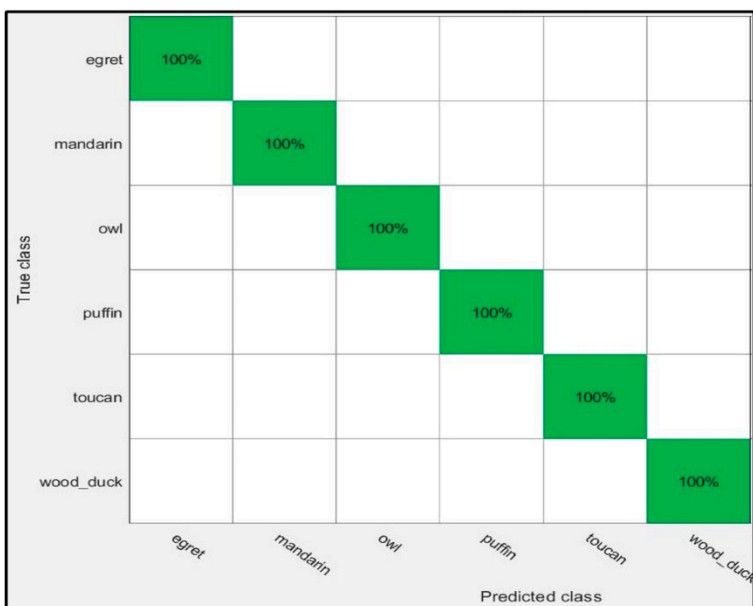

**Figure 8.** Confusion matrix for Birds dataset using proposed selection method on ESD classifier.

### 5.3. Butterflies Dataset

The results for the Butterflies dataset are given in Table 4. It may be observed that the ESD classifier gives better outcomes for all three feature methods. For M1, the ESD classifier achieves an accuracy of 95.1%, which is improved to 95.6% after using the P-Fusion method. The computational time of M1 is 46.05 (s), but after P-Fusion, the time is reduced to 31.95 (s). In comparison, the P-Selection method achieves an accuracy of 98%, which is better than the M1 and P-Fusion. Moreover, the computational time of this method is 19.53 (s), which is also the minimum. The performance of the ESD classifier for the P-Selection method may also be verified through Figure 9. The performance of the ESD classifier is also compared with a few other well-known techniques such as SVM, KNN, and LDA, as given in Table 4. From the results, it can be clearly seen that all the classifiers provide better accuracy on the P-Selection method. Moreover, it is also concluded that W-KNN performs better in terms of computational time.

**Table 4.** Proposed classification results using the Butterflies dataset. M1 represents simple serial-based fusion and classification, P-Fusion represents the proposed fusion approach, and P-Selection represents the proposed selection method results.

| Classifier | M1 | P-Fusion | P-Selection | Accuracy (%) | FNR (%) | Time (s) |
|---|---|---|---|---|---|---|
| | ✓ | - | - | **95.1** | 9.4 | 46.05 |
| ESD | - | ✓ | - | **95.6** | 5.9 | 31.95 |
| | - | - | ✓ | **98.0** | 2.0 | 19.53 |
| | ✓ | - | - | 85.7 | 14.3 | 28.56 |
| E-S-KNN | - | ✓ | - | 87.7 | 12.3 | 18.27 |
| | - | - | ✓ | 88.7 | 11.3 | 13.08 |

**Table 4.** *Cont.*

| Classifier | M1 | P-Fusion | P-Selection | Accuracy (%) | FNR (%) | Time (s) |
|---|---|---|---|---|---|---|
| | ✓ | - | - | 70.9 | 29.1 | 48.44 |
| LD | - | ✓ | - | 94.1 | 4.6 | 22.42 |
| | - | - | ✓ | 96.6 | 3.4 | 17.01 |
| | ✓ | - | - | 91.6 | 8.4 | 40.02 |
| L-SVM | - | ✓ | - | 94.6 | 5.4 | 29.65 |
| | - | - | ✓ | 96.6 | 3.4 | 16.72 |
| | ✓ | - | - | 94.1 | 5.9 | 39.46 |
| Q-SVM | - | ✓ | - | 94.1 | 5.9 | 24.58 |
| | - | - | ✓ | 96.6 | 3.4 | 18.80 |
| | ✓ | - | - | 90.6 | 4.9 | 44.23 |
| Cub-SVM | - | ✓ | - | 93.6 | 6.4 | 29.41 |
| | - | - | ✓ | 97.0 | 3.0 | 21.51 |
| | ✓ | - | - | 85.7 | 14.3 | 30.82 |
| F-KNN | - | ✓ | - | 89.2 | 10.8 | 18.70 |
| | - | - | ✓ | 94.1 | 5.9 | 13.79 |
| | ✓ | - | - | 82.3 | 19.7 | 29.29 |
| M-KNN | - | ✓ | - | 85.2 | 14.8 | 18.30 |
| | - | - | ✓ | 92.1 | 7.9 | 10.83 |
| | ✓ | - | - | 85.2 | 14.8 | **15.06** |
| W-KNN | - | ✓ | - | 87.2 | 12.8 | **14.26** |
| | - | - | ✓ | 94.6 | 5.4 | **10.12** |
| | ✓ | - | - | 81.8 | 18.2 | 16.02 |
| Cos-KNN | - | ✓ | - | 85.7 | 14.3 | 14.54 |
| | - | - | ✓ | 94.1 | 5.9 | 10.55 |

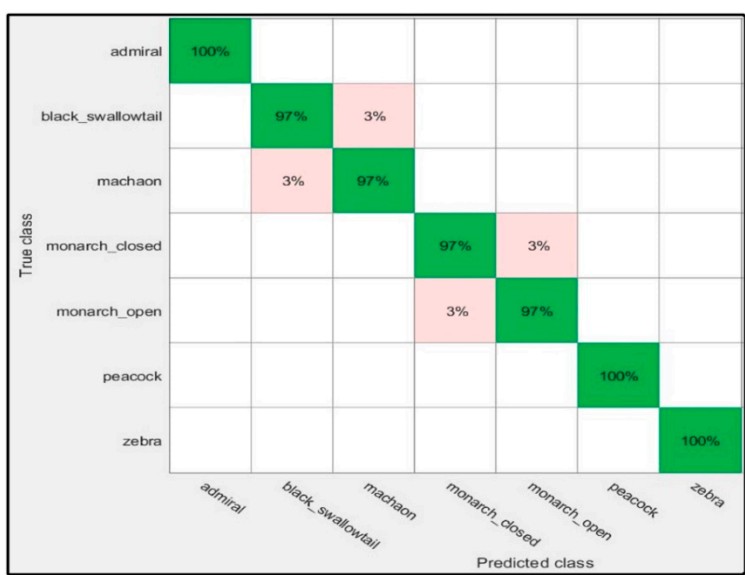

**Figure 9.** Confusion matrix for Butterflies dataset.

*5.4. CIFAR-100 Dataset*

This dataset consists of 100 object classes such as bus, chair, table, train, bed, and each class consists of 100 samples, making this dataset more challenging. There are 50,000 images available for the training of this dataset, while there are 10,000 images for testing. In this work, we utilize this dataset for the evaluation of the proposed technique. The results are given in Tables 4 and 5. In Table 4, the proposed training results are provided, which show the maximum accuracy of 69.76% and an error rate of 30.24%. For the simple fusion method (M1), the noted accuracy is 51.34%, and the computation time is 608 (min). After employing the proposed fusion, it takes the time of 524 (min) for execution, and achieved an improved accuracy of 63.97%. The proposed P-Selection method further improves the accuracy and reached 69.76%, whereas the execution time is also minimized to 374 (min). The testing results are given in Table 6. The maximum achieved accuracy of the testing process is 68.80% using the P-Selection method and ESD classifier. The accuracy is not impressive, but in the view of dataset complexity, it is acceptable. The accuracy of the ESD using the P-Selection method can be further verified through Figure 10 (confusion matrix).

**Table 5.** Proposed training results on CIFAR-100 dataset.

| Classifier | M1 | P-Fusion | P-Selection | Accuracy (%) | FNR (%) | Time (min) |
|---|---|---|---|---|---|---|
| | ✓ | - | - | 51.34 | 48.66 | 608 |
| **ESD** | - | ✓ | - | 63.97 | 36.03 | 524 |
| | - | - | ✓ | **69.76** | **30.24** | **374** |

**Table 6.** Proposed testing results on CIFAR-100 dataset.

| Classifier | M1 | P-Fusion | P-Selection | Accuracy (%) | FNR (%) | Time (min) |
|---|---|---|---|---|---|---|
| | ✓ | - | - | 47.84 | 52.16 | 258 |
| **ESD** | - | ✓ | - | 62.34 | 37.66 | 204 |
| | - | - | ✓ | **68.80** | **31.2** | **111** |

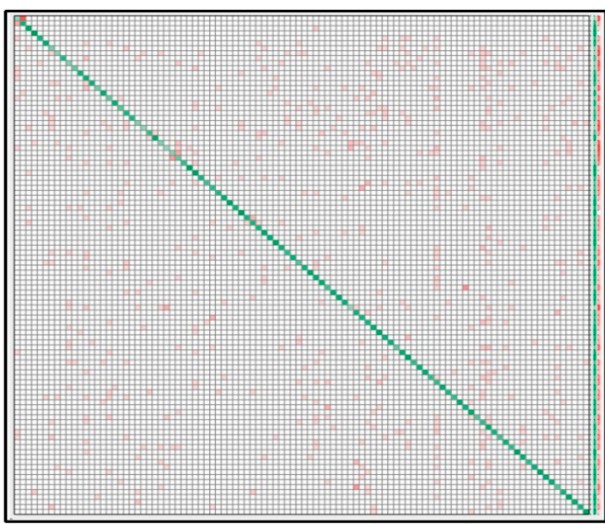

**Figure 10.** Confusion matrix of CIFAR-100 dataset for proposed P-Selection method.

*5.5. Analysis and Comparison with Existing Techniques*

A comprehensive analysis and comparison with existing techniques are presented in this section to examine the authenticity of the proposed method results. The proposed fusion and robust feature selection methods give a significant performance of 95.5%, 100%, 98%, and 68.70%, respectively, for ESD classifier on the selected datasets. Results can be seen in Tables 2–4. However, it is essential to examine the accuracy of ESD against each classifier based on a detailed statistical analysis. For Caltech-101

dataset, we run the proposed algorithm 500 times for each method and get two accuracies: average (76.3%, 87.9%, and 92.7%), and maximum (79%, 90.8%, and 95.5%). These accuracies are also plotted in Figure 11a. In this figure, it is shown that a minor change is occurring in the accuracy after 500 iterations. For the Birds database, two accuracies are also obtained: minimum (97.2%, 98.9%, and 99.4%) and maximum (99%, 99.5%, and 100%). These values are also plotted in Figure 11b. In this figure, it can be observed that the change in M1 is a bit higher as compared to P-Fusion and P-Selection. In the end, the statistical analysis is conducted for the Butterflies dataset, as shown in Figure 11c. This figure shows a slight change in the accuracy of each method.

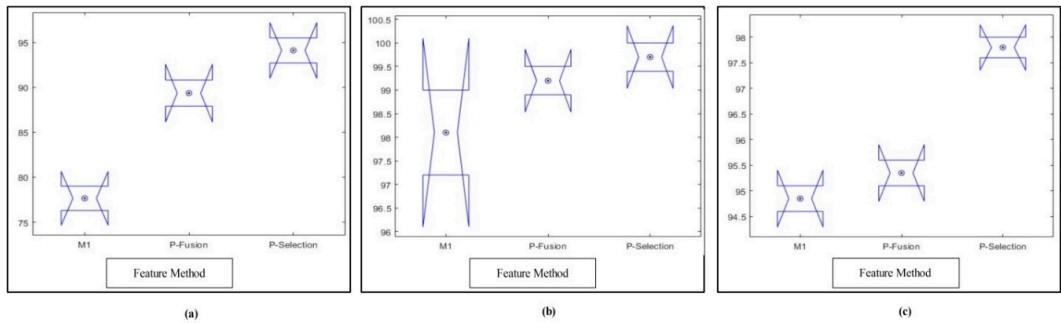

**Figure 11.** Statistical analysis of ESD classifier using all three methods. Where (**a**) represent M1 method, (**b**) denotes P-Fusion method, and (**c**) denotes P-Selection method, respectively.

We performed the classification using other deep neural nets such as VGG16, AlexNet, ResNet50, and ResNet101 to compare the proposed scheme classification performance. The results are computed from the last two layers, such as Vgg16 (FC7 and FC8), AlexNet (FC7 and FC8), and ResNet (Average Pool and FC Layer). The features extracted from these layers are fused using the proposed approach and later perform the selection technique. For the classification of these neural nets, we used the original classifier named Softmax. Results are given in Tables 7 and 8 below for Caltech-101 and CIFAR-100 datasets. In these tables, we noticed that the P-Fusion and P-Selection techniques are performed well using the proposed scheme. A brief comparison with existing techniques is also presented in Table 9. From this table, we computed the results on different training/testing ratios and get a variety of results. Based on the results, it is show that the increase in a training ratio minimizes the error rate. For example, in this table, accuracy of CIFAR-100 is 65.46%, 68.80%, 73.16%, and 77.28% for training/testing ratio 50:50, 60:40, 70:30, and 80:20, respectively. The minimum error rate is 22.72% for 80:30 approach whereas for standard approach (70:30), error rate is 26.84%. From this table, it is evident that the proposed method gives improved accuracy.

**Table 7.** Classification results on Caltech-101 dataset using different neural nets.

| Method | Features | | Measures | |
|---|---|---|---|---|
| | P-Fusion | P-Selection | Accuracy (%) | FNR (%) |
| AlexNet | ✓ | - | 86.70 | 13.30 |
| | - | ✓ | 90.24 | 9.76 |
| Vgg16 | ✓ | - | 85.16 | 14.84 |
| | - | ✓ | 89.24 | 10.76 |
| ResNet50 | ✓ | - | 88.57 | 11.43 |
| | - | ✓ | 92.36 | 7.64 |
| ResNet101 | ✓ | - | 89.96 | 10.04 |
| | - | ✓ | 92.83 | 7.17 |

**Table 7.** *Cont.*

| Method | Features | | Measures | |
|---|---|---|---|---|
| | **P-Fusion** | **P-Selection** | **Accuracy (%)** | **FNR (%)** |
| **Proposed** | ✓ | - | **90.80** | **9.20** |
| | - | ✓ | **95.50** | **4.50** |

**Table 8.** Classification results on CIFAR-100 dataset using different neural nets.

| Method | Features | | Measures | |
|---|---|---|---|---|
| | **P-Fusion** | **P-Selection** | **Accuracy (%)** | **FNR (%)** |
| AlexNet | ✓ | - | 61.29 | 38.71 |
| | - | ✓ | 65.82 | 34.18 |
| Vgg16 | ✓ | - | 60.90 | 39.10 |
| | - | ✓ | 64.06 | 35.94 |
| ResNet50 | ✓ | - | 61.82 | 38.18 |
| | - | ✓ | 65.71 | 34.29 |
| ResNet101 | ✓ | - | 61.98 | 38.02 |
| | - | ✓ | 66.25 | 33.75 |
| **Proposed** | ✓ | - | **62.34** | 38.71 |
| | - | ✓ | **68.80** | 34.18 |

**Table 9.** Comparison of proposed accuracy with recent techniques. MLFFS = Multi-Layers Features Fusion and Selection.

| Reference | Technique | Dataset | Accuracy (%) |
|---|---|---|---|
| Roshan et al. [25] | Fine-tuning on top layers | Caltech-101 | 91.66 |
| Jongbin et al. [26] | Discrete Fourier transform | Caltech-101 | 93.60 |
| Qun et al. [27] | Memory banks-based unsupervised learning | Caltech-101 | 91.00 |
| Qing et al. [28] | PCA-based reduction on fused features | Caltech-101 | 92.54 |
| Xueliang et al. [29] | A fusion of mid-level layers-based features | Caltech-101 | 92.20 |
| Rashid et al. [8] | Fusion of SIFT and CNN features | Caltech-101 | 89.70 |
| Svetlana [43] | Local affine parts-based approach | Butterflies | 90.40 |
| Ma et al. [44] | Genetic CNN designer approach (70:30) | CIFAR-100 | 66.77 |
| | IRRCNN (70:30) | CIFAR-100 | 72.78 |
| Alom et al. [45] | IRCNN (70:30) | CIFAR-100 | 71.76 |
| | EIN (70:30) | CIFAR-100 | 68.29 |
| | EIRN (70:30) | CIFAR-100 | 69.22 |
| Proposed | MLFFS | Butterflies | 98.00 |
| Proposed | MLFFS | Birds | 100% |
| Proposed | MLFFS | Caltech-101 | 95.5 |
| Proposed | MLFFS (50:50) | CIFAR-100 | 65.46 |
| - | MLFFS (60:40) | CIFAR-100 | 68.80 |
| - | MLFFS (70:30) | CIFAR-100 | 73.16 |
| - | MLFFS (80:20) | CIFAR-100 | 77.28 |

## 6. Conclusions

A new multi-layer deep features fusion and selection-based method for object classification is presented in this work. The major contribution of this work lies in the fusion of deep learning models, and then selection of the robust features for final classification. Three core steps are involved in the proposed system: Feature extraction using transfer learning, features fusion of two different deep learning models (VGG19 and Inception V3) using PMC, and selection of the robust features using

Multi Logistic Regression controlled Entropy-Variances (MRcEV) method. An ESDA classifier is used to validate the performance of MRcEV. We utilize three datasets for the experimental process and demonstrate an improved achieved accuracy. From the results, we conclude that the proposed method is useful for large, as well as small datasets. The fusion of two different deep learning features shows an impact on classification accuracy. Additionally, the selection of robust features shows an effect on both computational time and classification accuracy. The main limitation of the proposed method is the quality of features—by using low-quality images, it is not possible to get strong features. In the future, this problem will be rectified through contrast, stretching deep learning architecture. Moreover, for the improvement of experimental process, the Caltech-256 and CIFAR-100 datasets will be considered.

**Author Contributions:** M.R. and M.A.K. developed this idea, and they were responsible for the first draft. M.A. was responsible for mathematical formulation. S.-H.W. supervised this work. S.R.N. gave technical support for this work. T.S. and A.R. were responsible for the final proofreading. All authors have read and agreed to the published version of the manuscript.

**Funding:** There was no funding involved in this work.

**Conflicts of Interest:** The authors declare no conflict of interest.

## Appendix A

**Table A1.** Detailed description of VGG19 pre-trained CNN model.

| Sr No. | Name | Type | Activation | Learnable | | Total Learnables |
|---|---|---|---|---|---|---|
| | | | | **Weights** | **Bias** | |
| 1 | Input | Image Input | $224 \times 224 \times 3$ | - | - | - |
| 2 | conv1_1 | Convolution | $224 \times 224 \times 64$ | $3 \times 3 \times 3 \times 64$ | $1 \times 1 \times 64$ | 1792 |
| 3 | relu1_1 | ReLU | $224 \times 224 \times 64$ | - | - | - |
| 4 | conv1_2 | Convolution | $224 \times 224 \times 64$ | $3 \times 3 \times 64 \times 64$ | $1 \times 1 \times 64$ | 36,928 |
| 5 | relu1_2 | ReLU | $224 \times 224 \times 64$ | - | - | - |
| 6 | pool1 | Max Pooling | $112 \times 112 \times 64$ | - | - | - |
| 7 | conv2_1 | Convolution | $112 \times 112 \times 128$ | $3 \times 3 \times 64 \times 128$ | $1 \times 1 \times 128$ | 73,856 |
| 8 | relu2_1 | ReLU | $112 \times 112 \times 128$ | - | - | - |
| 9 | conv2_2 | Convolution | $112 \times 112 \times 128$ | $3 \times 3 \times 128 \times 128$ | $1 \times 1 \times 128$ | 147,584 |
| 10 | relu2_2 | ReLU | $112 \times 112 \times 128$ | - | - | - |
| 11 | pool2 | Max Pooling | $56 \times 56 \times 128$ | - | - | - |
| 12 | conv3_1 | Convolution | $56 \times 56 \times 256$ | $3 \times 3 \times 128 \times 256$ | $1 \times 1 \times 256$ | 295,168 |
| 13 | relu3_1 | ReLU | $56 \times 56 \times 256$ | - | - | - |
| 14 | conv3_2 | Convolution | $56 \times 56 \times 256$ | $3 \times 3 \times 256 \times 256$ | $1 \times 1 \times 256$ | 590,080 |
| 15 | relu3_2 | ReLU | $56 \times 56 \times 256$ | - | - | - |
| 16 | conv3_3 | Convolution | $56 \times 56 \times 256$ | $3 \times 3 \times 256 \times 256$ | $1 \times 1 \times 256$ | 590,080 |
| 17 | relu3_3 | ReLU | $56 \times 56 \times 256$ | - | - | - |
| 18 | conv3_4 | Convolution | $56 \times 56 \times 256$ | $3 \times 3 \times 256 \times 256$ | $1 \times 1 \times 256$ | 590,080 |
| 19 | relu3_4 | ReLU | $56 \times 56 \times 256$ | - | - | - |
| 20 | pool3 | Max Pooling | $28 \times 28 \times 256$ | - | - | - |
| 21 | conv4_1 | Convolution | $28 \times 28 \times 512$ | $3 \times 3 \times 256 \times 512$ | $1 \times 1 \times 512$ | 1,180,160 |
| 22 | relu4_1 | ReLU | $28 \times 28 \times 512$ | - | - | - |
| 23 | conv4_2 | Convolution | $28 \times 28 \times 512$ | $3 \times 3 \times 512 \times 512$ | $1 \times 1 \times 512$ | 2,359,808 |
| 24 | relu4_2 | ReLU | $28 \times 28 \times 512$ | - | - | - |
| 25 | conv4_3 | Convolution | $28 \times 28 \times 512$ | $3 \times 3 \times 512 \times 512$ | $1 \times 1 \times 512$ | 2,359,808 |
| 26 | relu4_3 | ReLU | $28 \times 28 \times 512$ | - | - | - |
| 27 | conv4_4 | Convolution | $28 \times 28 \times 512$ | $3 \times 3 \times 512 \times 512$ | $1 \times 1 \times 512$ | 2,359,808 |
| 28 | relu4_4 | ReLU | $28 \times 28 \times 512$ | - | - | - |
| 29 | pool4 | Max Pooling | $14 \times 14 \times 512$ | - | - | - |
| 30 | conv5_1 | Convolution | $14 \times 14 \times 512$ | $3 \times 3 \times 512 \times 512$ | $1 \times 1 \times 512$ | 2,359,808 |
| 31 | relu5_1 | ReLU | $14 \times 14 \times 512$ | - | - | - |
| 32 | conv5_2 | Convolution | $14 \times 14 \times 512$ | $3 \times 3 \times 512 \times 512$ | $1 \times 1 \times 512$ | 2,359,808 |
| 33 | relu5_2 | ReLU | $14 \times 14 \times 512$ | - | - | - |

**Table A1.** *Cont.*

| Sr No. | Name | Type | Activation | Learnable | | Total Learnables |
|---|---|---|---|---|---|---|
| | | | | **Weights** | **Bias** | |
| 34 | conv5_3 | Convolution | $14 \times 14 \times 512$ | $3 \times 3 \times 512 \times 512$ | $1 \times 1 \times 512$ | 2,359,808 |
| 35 | relu5_3 | ReLU | $14 \times 14 \times 512$ | - | - | - |
| 36 | conv5_4 | Convolution | $14 \times 14 \times 512$ | $3 \times 3 \times 512 \times 512$ | $1 \times 1 \times 512$ | 2,359,808 |
| 37 | relu5_4 | ReLU | $14 \times 14 \times 512$ | - | - | - |
| 38 | pool5 | Max Pooling | $7 \times 7 \times 512$ | - | - | - |
| 39 | fc6 | Fully Connected | $1 \times 1 \times 4096$ | $4096 \times 25,088$ | $4096 \times 1$ | 102,764,544 |
| 40 | relu6 | ReLU | $1 \times 1 \times 4096$ | - | - | - |
| 41 | drop6 | Dropout | $1 \times 1 \times 4096$ | - | - | - |
| 42 | fc7 | Fully Connected | $1 \times 1 \times 4096$ | $4096 \times 4096$ | $4096 \times 1$ | 16,781,312 |
| 43 | relu7 | ReLU | $1 \times 1 \times 4096$ | - | - | - |
| 44 | drop7 | Dropout | $1 \times 1 \times 4096$ | - | - | - |
| 45 | fc8 | Fully Connected | $1 \times 1 \times 1000$ | $1000 \times 4096$ | $1000 \times 1$ | 4,097,000 |
| 46 | Prob | Softmax | $1 \times 1 \times 1000$ | - | - | - |
| 47 | Output | Classification | | - | - | - |

**Table A2.** Detailed description of Inception V3 pre-trained CNN model.

| S/N | Name | Type | Activation | Learnable | | | |
|---|---|---|---|---|---|---|---|
| | | | | **Weights** | **Bias** | **Offset** | **Scale** |
| 1 | input_1 | Image Input | $299 \times 299 \times 3$ | - | - | - | - |
| 2 | scaling | Scaling | $299 \times 299 \times 3$ | - | - | - | - |
| 3 | conv2d_1 | Convolution | $149 \times 149 \times 32$ | [3,3,3,32] | [1,1,32] | - | - |
| 4 | batch_normalization_1 | Batch Normalization | $149 \times 149 \times 32$ | - | - | $1 \times 1 \times 32$ | $1 \times 1 \times 32$ |
| 5 | activation_1_relu | ReLU | $149 \times 149 \times 32$ | - | - | - | - |
| 6 | conv2d_2 | Convolution | $147 \times 147 \times 32$ | [3,3,32,32] | [1,1,32] | - | - |
| 7 | batch_normalization_2 | Batch Normalization | $147 \times 147 \times 32$ | - | - | [1,1,32] | [1,1,32] |
| 8 | activation_2_relu | ReLU | $147 \times 147 \times 32$ | - | - | - | - |
| 9 | conv2d_3 | Convolution | $147 \times 147 \times 64$ | [3,3,32,64] | [1,1,64] | - | - |
| 10 | batch_normalization_3 | Batch Normalization | $147 \times 147 \times 64$ | - | - | [1,1,64] | [1,1,64] |
| 11 | activation_3_relu | ReLU | $147 \times 147 \times 64$ | - | - | - | - |
| 12 | max_pooling2d_1 | Max Pooling | $73 \times 73 \times 64$ | - | - | - | - |
| 13 | conv2d_4 | Convolution | $73 \times 73 \times 80$ | [1,1,64,80] | [1,1,80] | - | - |
| 14 | batch_normalization_4 | Batch Normalization | $73 \times 73 \times 80$ | - | - | [1,1,80] | [1,1,80] |
| 15 | activation_4_relu | ReLU | $73 \times 73 \times 80$ | - | - | - | - |
| 16 | conv2d_5 | Convolution | $71 \times 71 \times 192$ | [3,3,80,192] | [1,1,192] | - | - |
| 17 | batch_normalization_5 | Batch Normalization | $71 \times 71 \times 192$ | - | - | [1,1,192] | [1,1,192] |
| 18 | activation_5_relu | ReLU | $71 \times 71 \times 192$ | - | - | - | - |
| 19 | max_pooling2d_2 | Max Pooling | $35 \times 35 \times 192$ | - | - | - | - |
| 20 | conv2d_9 | Convolution | $35 \times 35 \times 64$ | [1,1,192,64] | [1,1,64] | - | - |
| 21 | batch_normalization_9 | Batch Normalization | $35 \times 35 \times 64$ | - | - | [1,1,64] | [1,1,64] |
| 22 | activation_9_relu | ReLU | $35 \times 35 \times 64$ | - | - | - | - |
| 23 | conv2d_7 | Convolution | $35 \times 35 \times 48$ | [1,1,192,48] | [1,1,48] | - | - |
| 24 | conv2d_10 | Convolution | $35 \times 35 \times 96$ | [3,3,64,96] | [1,1,96] | - | - |
| 25 | batch_normalization_7 | Batch Normalization | $35 \times 35 \times 48$ | - | - | [1,1,48] | [1,1,48] |
| 26 | batch_normalization_10 | Batch Normalization | $35 \times 35 \times 96$ | - | - | [1,1,96] | [1,1,96] |
| 27 | activation_7_relu | ReLU | $35 \times 35 \times 48$ | - | - | - | - |
| 28 | activation_10_relu | ReLU | $35 \times 35 \times 96$ | - | - | - | - |
| 29 | average_pooling2d_1 | Avg Pooling | $35 \times 35 \times 192$ | - | - | - | - |
| 30 | conv2d_6 | Convolution | $35 \times 35 \times 64$ | [1,1,192,64] | [1,1,64] | - | - |
| 31 | conv2d_8 | Convolution | $35 \times 35 \times 64$ | [5,5,48,64] | [1,1,64] | - | - |
| 32 | conv2d_11 | Convolution | $35 \times 35 \times 92$ | [3,3,96,96] | [1,1,96] | - | - |
| 33 | conv2d_12 | Convolution | $35 \times 35 \times 32$ | [1,1,192,32] | [1,1,32] | - | - |
| 34 | batch_normalization_6 | Batch Normalization | $35 \times 35 \times 64$ | - | - | [1,1,64] | [1,1,64] |
| 35 | batch_normalization_8 | Batch Normalization | $35 \times 35 \times 64$ | - | - | [1,1,64] | [1,1,64] |
| 36 | batch_normalization_11 | Batch Normalization | $35 \times 35 \times 96$ | - | - | [1,1,96] | [1,1,96] |
| 37 | batch_normalization_12 | Batch Normalization | $35 \times 35 \times 32$ | - | - | [1,1,32] | [1,1,32] |
| 38 | activation_6_relu | ReLU | $35 \times 35 \times 64$ | - | - | - | - |
| 39 | activation_8_relu | ReLU | $35 \times 35 \times 64$ | - | - | - | - |
| 40 | activation_11_relu | ReLU | $35 \times 35 \times 96$ | - | - | - | - |
| 41 | activation_12_relu | ReLU | $35 \times 35 \times 32$ | - | - | - | - |
| 42 | mixed0 | Depth Concat | $35 \times 35 \times 256$ | - | - | - | - |
| 43 | conv2d_16 | Convolution | $35 \times 35 \times 64$ | [1,1,256,64] | [1,1,64] | - | - |
| 44 | batch_normalization_16 | Batch Normalization | $35 \times 35 \times 64$ | - | - | [1,1,64] | [1,1,64] |
| 45 | activation_16_relu | Fully Connected | $35 \times 35 \times 64$ | - | - | - | - |
| 46 | conv2d_14 | Convolution | $35 \times 35 \times 48$ | [1,1,256,48] | [1,1,48] | - | - |

**Table A2.** *Cont.*

| S/N | Name | Type | Activation | Learnable | | | |
|-----|------|------|------------|-----------|---|---|---|
| | | | | Weights | Bias | Offset | Scale |
| 47 | conv2d_17 | Convolution | $35 \times 35 \times 96$ | [3,3,64,96] | [1,1,96] | - | - |
| – | – | – | – | – | – | – | – |
| 307 | batch_normalization_94 | Batch Normalization | $8 \times 8 \times 192$ | - | - | [1,1,192] | [1,1,192] |
| 308 | activation_86_relu | ReLU | $8 \times 8 \times 320$ | - | - | - | - |
| 309 | mixed9_1 | Depth Concat | $8 \times 8 \times 768$ | - | - | - | - |
| 310 | concatenate_2 | Depth Concat | $8 \times 8 \times 768$ | - | - | - | - |
| 311 | activation_94_relu | ReLU | $8 \times 8 \times 192$ | - | - | - | - |
| 312 | mixed10 | Depth Concat | $8 \times 8 \times 2048$ | - | - | - | - |
| 313 | avg_pool | Avg Pooling | $1 \times 1 \times 2048$ | - | - | - | - |
| 314 | predictions | Fully Connected | $1 \times 1 \times 1000$ | $1000 \times 2048$ | $1000 \times 1$ | - | - |
| 315 | predictions_softmax | Softmax | $1 \times 1 \times 1000$ | - | - | - | - |
| 316 | classification layer_predictions | Classification Output | | - | - | - | - |

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
