# Peer review of "A Sustainable Deep Learning Framework for Object Recognition Using Multi-Layers Deep Features Fusion and Selection"

_sustainability, doi:10.3390/su12125037_

Round 1
Reviewer 1 Report
This paper proposed a three-step approach to improve object classification. Experiments show that P-fusion and P-selection can improve accuracy, and decrease computational time on multiple types of classifiers and three datasets.
I have a few questions:
(1) Why do you choose the three datasets (the Birds and Butterfly seems to be quite small and easy)? Why not choose some more "standard" datasets such as CIFAR100 and ImageNet? If you can demonstrate that your method improve accuracy over baseline on large dataset as ImageNet, it would be much more convincing.
(2) Also, apart from ESD, the other main classifiers used are SVM and KNN. What would happen if you take some other deep neural nets on the leaderboard and see if your how the P-fusion and P-selection may improve the accuracy?
(3) Why do you use Parallel Maximum Covariance (PMC) instead of CCA? What's PMC's advantage?
Some other questions or comments:
(4) What does "cd" mean in Eq. 20?
(5) There is no need to put the Table 1 and 2 in the main text, since they are not the main contribution. It can be put in the appendix.
(6) Missing reference in Line 75 and 163.
(7) The English needs polished. And needs much more clear explanation in 4.3 Feature Selection (including justification)
Author Response
Comments and Suggestions for Authors:
This paper proposed a three-step approach to improve object classification. Experiments show that P-fusion and P-selection can improve accuracy, and decrease computational time on multiple types of classifiers and three datasets.
I have a few questions:
Comment 1: Why do you choose the three datasets (the Birds and Butterfly seems to be quite small and easy)? Why not choose some more "standard" datasets such as CIFAR100 and ImageNet? If you can demonstrate that your method improve accuracy over baseline on large dataset as ImageNet, it would be much more convincing.
Response: As recommended by the honorable reviewer, we have now evaluated our proposed method on the CIFAR-100 dataset, and incorporated the obtained results within the revised manuscript. The results are obtained for both training process as well as testing process. You will be pleased to see that the proposed framework manages to achieve better accuracy than the existing literature.
Compared to Caltech101, this dataset is not more complex due to the following reason: Caltech101 dataset also includes 101 object classes, and each class does not only carry static images like CIFAR-100, in which each class contains 100 images. This makes Caltech101 imbalanced. The other datasets like Birds and Butterfly are utilized to check the performance of the proposed technique on smaller datasets.
Comment 2: Also, apart from ESD, the other main classifiers used are SVM and KNN. What would happen if you take some other deep neural nets on the leaderboard and see if your how the P-fusion and P-selection may improve the accuracy?
Response: As per recommendation by honorable reviewer, we perform the classification using other deep neural nets such as VGG16, AlexNet, ResNet50, and ResNet101 to compare the proposed scheme classification performance. The results are computed from last two layers such as Vgg16 (FC7 and FC8), AlexNet (FC7 and FC8), and ResNet (Average Pool and FC Layer). The features extracted from these layers are fused using proposed approach and later perform selection technique. For classification of these neural nets, we used original classifier named Softmax. Results are given in Tables below for Caltech101 and CIFAR100 datasets. In these tables, we noticed that the P-Fusion and P-Selection techniques are performed well using proposed scheme. These results are also described in the revised manuscript.
Table 1: Classification results on Caltech101 dataset using different neural nets
|
Method |
Features |
Measures |
||
|
P-Fusion |
P-Selection |
Accuracy (%) |
FNR (%) |
|
|
AlexNet |
ü |
|
86.70 |
13.30 |
|
|
ü |
90.24 |
9.76 |
|
|
Vgg16 |
ü |
|
85.16 |
14.84 |
|
|
ü |
89.24 |
10.76 |
|
|
ResNet50 |
ü |
|
88.57 |
11.43 |
|
|
ü |
92.36 |
7.64 |
|
|
ResNet101 |
ü |
|
89.96 |
10.04 |
|
|
ü |
92.83 |
7.17 |
|
|
Proposed |
ü |
|
90.80 |
9.20 |
|
|
ü |
95.50 |
4.50 |
|
Table 2: Classification results on CIFAR100 dataset using different neural nets
|
Method |
Features |
Measures |
||
|
P-Fusion |
P-Selection |
Accuracy (%) |
FNR (%) |
|
|
AlexNet |
ü |
|
61.29 |
38.71 |
|
|
ü |
65.82 |
34.18 |
|
|
Vgg16 |
ü |
|
60.90 |
39.10 |
|
|
ü |
64.06 |
35.94 |
|
|
ResNet50 |
ü |
|
61.82 |
38.18 |
|
|
ü |
65.71 |
34.29 |
|
|
ResNet101 |
ü |
|
61.98 |
38.02 |
|
|
ü |
66.25 |
33.75 |
|
|
Proposed |
ü |
|
62.34 |
38.71 |
|
|
ü |
68.80 |
34.18 |
|
Comment 3: Why do you use Parallel Maximum Covariance (PMC) instead of CCA? What's PMC's advantage?
Response: Features fusion based on canonical correlation analysis (CCA) utilize the correlation among two feature sets to compute two sets of transformed features. The transformed features have higher correlation than the two feature sets. The main limitation of CCA is that it ignores the relation between class structures among the complex image dataset like Caltech101. Whereas, we are interested in maximizing the correlation among feature sets and separating the classes within each set of features. So the main advantage of the proposed PMC approach is that it maximizes the correlation among features and also the difference between object classes.
Some other questions or comments:
Comment 4: What does "cd" mean in Eq. 20?
Response: Thank you for pointing this out. This equation has been modified in the revised version.
Comment 5: There is no need to put the Table 1 and 2 in the main text, since they are not the main contribution. It can be put in the appendix.
Response: As per the recommendation, the mentioned tables are included in the Appendix.
Comment 6: Missing reference in Line 75 and 163.
Response: Both the references have been corrected.
Comment 7: The English needs polished. And needs much more clear explanation in 4.3 Feature Selection (including justification)
Response: The entire manuscript has been thoroughly proofread. We hope that you will find our corrections satisfactory.
Reviewer 2 Report
I am not expert for this kind of field, but study, methods and manuscript are well written.
In Abstract all abbreviations should be in full name, eg. VGG19.
Figure 4,5,6 - can you use different background color for textbox?
Figure 1 --> line 407? Larger font.
Please check English
Author Response
Comments and Suggestions for Authors
I am not expert for this kind of field, but study, methods and manuscript are well written.
Comment: In Abstract all abbreviations should be in full name, eg. VGG19.
Response: As per the recommendation, all abbreviations are defined in the revised manuscript.
Comment: Figure 4,5,6 - can you use different background color for textbox?
Response: The background color of textboxes in Figures 4, 5, and 6 are updated in the revised version.
Comment: Figure 1 --> line 407? Larger font.
Response: As per recommendation, the font of Figure 10 has been updated in the revised manuscript.
Comment: Please check English
Response: As per recommendation, the whole manuscript is carefully proofread in terms of English language (grammar and Typos).

Round 2
Reviewer 1 Report
The authors address most of my comments. Thanks for making the changes and improving the paper.
For the added experiment of CIFAR100, the authors compare with several baselines such as VGG16 and ResNet50. However, the reported accuracy is much lower than achieved by some other repositories. For example, in this repository, https://github.com/weiaicunzai/pytorch-cifar100, they achieved 100-22.61=77.39% top-1 accuracy with ResNet50, while the authors report around 65\%. And the repository achieved 100-27.07=72.93% accuracy with VGG16, while the authors get around 64%. Why is the accuracy much lower?
Author Response
Response is attached

Round 3
Reviewer 1 Report
The authors addressed my concerns. I think the paper is good for publication.
Minor:
Line 428: "80:30" -> "80:20"